# Updates in Culture-Negative Endocarditis

**DOI:** 10.3390/pathogens12081027

**Published:** 2023-08-10

**Authors:** Jack McHugh, Omar Abu Saleh

**Affiliations:** Division of Public Health, Infectious Diseases, Occupational Medicine, Mayo Clinic, Rochester, MN 55901, USA; abusaleh.omar@mayo.edu

**Keywords:** endocarditis, culture-negative, bacteremia

## Abstract

Blood culture-negative infective endocarditis (BCNE) is a challenging condition associated with significant morbidity and mortality. This review discusses the epidemiology, microbiology, diagnosis, and treatment of BCNE considering advancements in molecular diagnostics and increased access to cardiac surgery. BCNE can be categorized into bacterial endocarditis with sterilized blood cultures due to previous antibiotic treatment, endocarditis caused by fastidious microorganisms, and true BCNE caused by intracellular organisms that cannot be cultured using traditional techniques. Non-infectious causes such as nonbacterial thrombotic endocarditis should also be considered. Diagnostic approaches involve thorough patient history; blood and serum testing, including appropriate handling of blood cultures; serological testing; and molecular techniques such as targeted and shotgun metagenomic sequencing. Where available, evaluation of explanted cardiac tissue through histopathology and molecular techniques is crucial. The therapy for BCNE depends on the likely causative agent and the presence of prosthetic material, with surgical intervention often required.

## 1. Introduction

Despite advances in antimicrobial diagnostics and therapeutics in the 20th century, mortality from infective endocarditis (IE) remains high. Over one-third of patients affected die within one year of diagnosis [1]. Blood cultures, allied with data obtained from clinical examination and echocardiography, are the gold standard for the diagnosis of IE. This is reflected in the 2023 Duke-International Society for Cardiovascular Infectious Diseases (ISCVID) IE criteria [2]. Conventional techniques for culturing blood do not always yield an organism, however. As per the Duke-ISCVID guidelines, a diagnosis of IE can still be established, provided there is sufficient supporting data. This subset of cases, hereafter referred to as blood culture-negative infective endocarditis (BCNE), presents a particular challenge for the clinician, and is associated with increased mortality.

The advent of non-culture-based diagnostics and increased access to cardiac surgery over the past two decades have allowed for the identification of culprit organisms in cases of IE that would previously have been considered culture-negative. This has led to changes in our understanding of the epidemiology, microbiology, treatment, and prognosis of BCNE. Accounting for contemporary advances, in this review we outline an updated approach to the diagnosis and treatment of BCNE.

## 2. Definition

Blood culture-negative endocarditis is defined as endocarditis where traditional methods for culturing blood do not yield an organism. BCNE may be classified into three groups [3]:
Bacterial endocarditis with sterilized blood cultures from previous antibiotic treatment, which accounts for the majority of cases;Endocarditis due to fastidious microorganisms, which have historically included the HACEK (*Haemophilus*, *Aggregatibacter*, *Cardiobacterium*, *Eikenella, Kingella*) group, nutritionally variant *Streptococci, Pasteurella* spp., mycobacteria, and fungal organisms;“True” BCNE due to infection with intra-cellular organisms that cannot be cultured in blood using traditional techniques but may be diagnosed with serology (e.g., *Bartonella* sp., *Coxiella burnetii*) or polymerase chain reaction (PCR) of valvular tissue (e.g., *Tropheryma whipplei*).

Non-infectious causes of endocarditis, specifically nonbacterial thrombotic endocarditis (NBTE), should also be included in the differential diagnosis of BCNE in the right clinical context. NBTE is associated with hypercoagulable states, e.g., malignancy or autoimmune disease, which may result in the deposition of sterile thrombi on cardiac valves.

## 3. Epidemiology of BCNE

A systematic review of 142 studies reporting IE microbiology over five decades from 1960–2010 found a decrease in the incidence of BCNE over that time period from 23.1% to 14.2% [4]. Conversely, Vogkou et al. reviewed 105 studies from 2003 to 2013, with 26.6% of cases classified as BCNE [5]. Accounting for these variations, the incidence of BCNE in modern times is likely in the range of 10–20% [6]. A notable feature of these reviews is the significant variation in the incidence of BCNE in individual studies, with rates ranging from 7.7% to 66% [7,8]. The variation in incidence is attributable to several factors, most notably local variation in the early use of antibiotic therapy prior to obtaining blood cultures but also differences in testing strategies [9] and geographic variation of specific organisms, in particular zoonotic agents [6].

## 4. Infectious and Non-Infectious Etiologies of BCNE

### 4.1. BCNE Associated with Previous Antibiotic Treatment

Sepsis is a common clinical presentation of IE, and as a result antibiotics are often administered prior to obtaining blood cultures. In retrospective studies, the prevalence of BCNE attributed to antibiotic administration prior to culture has varied from 35% to 74% [7,10,11]. The microbiological profile in this cohort is similar to that of cases where blood cultures are obtained prior to administration of antibiotics (i.e., streptococci, staphylococci, or enterococci), with the caveat that organisms that are quickly cleared from the bloodstream may be more common.

### 4.2. BCNE Associated with Fastidious Microorganisms

#### 4.2.1. The HACEK Group and Nutritionally Variant Streptococci

The HACEK group of organisms and nutritionally variant streptococci (*Granulicatella* spp., *Abiotrophia defectiva*) have traditionally been cited as common causes of BCNE due to the difficulty associated with culturing in standard media. However, modern automated culture systems can identify these organisms within 5 days of incubation in the vast majority of cases [12,13].

#### 4.2.2. Fungi

Fungi account for approximately 1–2% of cases of IE [6]. *Candida albicans* is the most common cause, accounting for around 25% of cases, and automated culture systems can readily culture this organism along with other yeasts. Invasive mold infections (e.g., IE associated with *Aspergillus* spp.) and IE associated with endemic fungi, e.g., *Histoplasma capsulatum*, do not traditionally grow on routine blood cultures, and close attention to risk factors in the history (Table 1) are important when considering these agents as potential etiologies of BCNE.

#### 4.2.3. Mycobacteria

IE due to *Mycobacterial* species is most frequently caused by non-tuberculous mycobacteria, with rapidly growing mycobacteria most commonly implicated [20,21]. Blood cultures do not routinely identify *Mycobacterial* species, and alternative diagnostic methodologies such as histopathologic evaluation of the valve and molecular techniques may be required to establish a diagnosis. IE due to *M. tuberculosis* is rare, with only a handful of cases reported in the literature. Cardiac surgery, the presence of indwelling prosthetic material, and intravenous drug use have been identified as the most common risk factors for mycobacterial endocarditis [21]. Several outbreaks of *M. chimaera* prosthetic valve endocarditis have been reported in the United States and Europe over the past decade [19]. These outbreaks have been associated with contaminated heater–cooler units during cardiopulmonary bypass, and this organism should be considered in patients presenting with prosthetic valve endocarditis of insidious onset. The significant lag time from the index surgery and subtle cardiac imaging findings contribute to the significant delay in diagnosis. Mycobacterial blood cultures and molecular detection methods should be performed in suspected cases.

#### 4.2.4. *Tropheryma whipplei*

*Tropheryma whipplei* is a Gram-positive bacterium that causes Whipple’s disease; a chronic multi-systemic disease that predominantly affects middle-aged men and typically causes diarrhea, weight loss, and arthralgias and that may progress to involve the heart, lungs, and central nervous system [22]. Estimates of the frequency of *T. whipplei* as a causative organism in BCNE have ranged from 0.3% to 3.5% [23,24,25] in studies examining all patients with BCNE and up to 6.3% in patients in whom cardiac tissue was available for analysis [26].

### 4.3. BCNE Associated with Intracellular Pathogens

#### 4.3.1. *Coxiella burnetii*

*Coxiella burnetii* is an obligate intracellular pathogen found worldwide that is associated with exposure to aerosols from infected animals (Table 1). Infection with this organism causes Q fever, a febrile illness associated with myalgias, headaches, and hepatitis and that may progress to endocarditis in the chronic phase if not diagnosed and treated early in the disease course. In the largest prospective studies evaluating cases of BCNE, *C. burnetii* was identified as the causative pathogen in 37% [24] and 48% [23] of cases. Positive phase 1 IgG titer is considered a major Duke criterion and is considered diagnostic. Coxiella specific PCR performed on the plasma or heart valves can also be used to confirm the diagnosis.

#### 4.3.2. *Bartonella* Species

*Bartonella* spp. are small intracellular Gram-negative bacteria with protean manifestations and have been associated with up to 12.4–28.4% of cases of BCNE [23,24]. *Bartonella henselae* is associated with cat-scratch disease, and clinicians should be aware of its association with crescentic glomerulonephritis with PR3-ANCA positivity [27]. *B. quintana* is associated with exposure to the human body louse (Table 1) and has also been associated with BCNE, along with rarer species such as *B. elizabethae* and *B. vinsonii* [28,29]. Although most of the literature with respect to *B. quintana* endocarditis has come out of Europe and Africa, a resurgence in North America has been noted in the past decade [30]. Bartonella serologies are the most commonly used diagnostic tests; Bartonella-specific PCR on peripheral blood specimens or tissue valves could also be used to confirm the diagnosis.

#### 4.3.3. Other Intracellular Pathogens

Rare cases of BCNE associated with *Legionella* spp., *Chlamydia* spp., and *Mycoplasma* spp. have been reported in the literature [18,31,32]. Cases of Legionella endocarditis are typically preceded by pneumonia. Diagnosis may be established with histopathological examination of explanted valves or molecular techniques; the role of serological testing is questionable given the rarity of these pathogens [33]. Brucellosis should also be considered in a patient with specific risk factors or returning from an endemic region (Table 1). Although serological testing for *Brucella* sp. can be added in these cases [34], detection in routine blood cultures is typically achieved within 5 days [35].

### 4.4. Non-Infectious Causes of IE

In the largest prospective evaluation of cases of BCNE at a tertiary reference center in France, Fournier et al. found a non-infectious cause of BCNE in 2.5% of cases [23]. Specific diagnoses included systemic lupus erythematosus (SLE), also known as Libman–Sacks endocarditis; rheumatoid arthritis; and IE associated with metastatic malignancy, also known as marantic endocarditis. Endocarditis has also been associated with Behçet’s disease [36]. Clinicians should also be aware of the link between endocarditis on porcine bioprosthetic valves and allergy to pork. A handful of cases of recurrent endocarditis on porcine valves in patients with pork allergies has been reported to date in the literature [37], and replacement of the valve with non-porcine material is necessary in these cases.

## 5. Diagnostic Approach

A thorough history should be obtained in all patients with IE to determine whether antibiotics were administered at home or elsewhere prior to obtaining blood cultures and to elicit risk factors for specific microorganisms, as outlined in Table 1. A comprehensive examination should also be performed to gather data on extra-cardiac symptoms that may be suggestive of a specific infectious etiology (e.g., joint and neurological involvement in Whipple’s disease) or non-infectious etiology (e.g., oral and genital ulceration in Behçet’s disease).

### 5.1. Blood and Serum Testing

#### 5.1.1. Appropriate Handling of Blood Cultures

In all cases of suspected IE, three sets of blood cultures, with each set containing one aerobic and one anaerobic bottle, should be obtained from different venipuncture sites, with the first and last samples drawn at least 1 h apart [1]. The yield of blood cultures is directly related to the volume of blood cultured, and it is therefore essential that bottles be adequately filled (e.g., 10 mL of blood per Bactec or BacT/Alert bottle). The HACEK organisms, nutritionally variant streptococci, and *Candida* sp. were traditionally considered challenging to detect in blood cultures, and prolonged incubation times were recommended. Thankfully, this is no longer the case with modern automated blood culture systems, with which these organisms are easily detected [13]. However, in situations in which blood cultures show no signs of infection after 5 days, it is still advisable to continue incubation for up to 14 days. Recent data suggest that this prolonged period may be beneficial for identifying *Cutibacterium* species [38]. Blind subcultures and terminal culture to chocolate agar at 5 days have also been recommended by the Clinical and Laboratory Standards Institute, but multiple studies have failed to demonstrate the utility of this approach [12,38].

#### 5.1.2. The Role of Serologic Testing

When the medical history or clinical presentation is suggestive of a zoonotic etiology, or when blood cultures are negative at 48 h, serological testing should be performed. *Coxiella burnetii* antiphase I immunoglobulin G (IgG) antibody titer ≥ 1:800 and indirect immunofluorescence assays (IFA) for the detection of IgM and IgG antibodies to *Bartonella henselae* or *Bartonella quintana* with IgG titer ≥ 1:800 are major microbiologic criteria for the diagnosis of IE in the 2023 modified Duke criteria [2], and both should be obtained in these cases. In the largest series examining BCNE to date, serological analysis provided a diagnosis for 356/745 (47.8%) patients with negative blood cultures; Q fever and bartonellosis accounted for 354 of these cases [23]. Clinicians should be aware that there is cross-reactivity between *Bartonella* spp. and *Coxiella burnetii*, although antibody titers against the true infecting agent tend to be higher [33]. In regions where *Brucella* is endemic, or where the patient has specific risk factors (Table 1), serologic testing for *Brucella* spp. should also be obtained. Serologic testing for extremely rare causes of endocarditis, e.g., *Chlamydia/Chlamydophila* species and *Legionella* species, is not routinely recommended due to challenges associated with false positive results, which may occur if the patient has had prior exposure to these organisms. In cases where there is a concern for rheumatologic disease, clinicians should also consider additional autoantibody serological testing with antinuclear antibody (ANA), rheumatoid factor (RF), and anti-double-stranded DNA.

#### 5.1.3. Molecular Techniques

Specific molecular methods that may aid in the diagnosis of IE include organism-specific PCR testing; broad-range PCR for the amplification of bacterial genetic material from 16S rRNA, also referred to as targeted metagenomic sequencing (tMGS); and shotgun metagenomic sequencing (sMGS), where all sequences of genomic DNA from a blood or tissue specimen are sampled. Although the sensitivity and specificity of these techniques are higher in explanted tissue over blood or plasma [33], cardiac tissue is only available in cases where surgery is indicated.

Organism-specific PCR testing has been developed for *C. burnetii*, *Bartonella* spp., and *T. whipplei*, among others. In the case of *C. burnetii* and *Bartonella* spp., PCR testing from plasma or whole blood is unlikely to add additional diagnostic benefit beyond that obtained by serologic testing [23]. *T. whipplei* PCR from peripheral blood was found to be positive in only 5 of 16 (31.2%) patients with Whipple’s endocarditis confirmed on a histology or PCR of valve tissue in one cases series [39] but should be considered in cases of subacute endocarditis where cultures and serology remain negative and surgery is not planned.

To date, one prospective cohort study has evaluated tMGS on plasma and whole blood [40]. In this study at a tertiary reference center, a pathogen was identified in five of six cases of BCNE. Results of this assay are typically available within 24–48 h. Limitations of tMGS include high cost, the inability to differentiate between viable and dead bacteria, the inability to predict anti-microbial resistance, limits on the number of bacteria available for detection, and the inability to detect non-bacterial organisms e.g., *Candida albicans*. Two small studies have evaluated sMGS using plasma microbial cell-free DNA (mcfDNA) testing [41,42]. Although no cases of BCNE were evaluated in these studies, the sensitivity of mcfDNA was similar to that of blood cultures, and mcfDNA was noted to remain present in blood samples for up to one month following initial infection, suggesting a role in cases where blood cultures have been sterilized by prior administration of antibiotics. Similar to tMGS, caution is needed with result interpretation, as sMGS does not differentiate between viable and dead bacteria. Potential advantages of sMGS are the ability to identify various microorganisms, including bacteria, fungi, and parasites, and the potential to detect antimicrobial resistance genes.

### 5.2. Evaluation of Explanted Cardiac Tissue

Surgical intervention is performed in approximately 22.5–51.2% [43,44] of patients with IE, and in all cases, prosthetic tissue and/or cardiac device material should be submitted for microbiologic and histopathologic study. Examination of valve tissue assumes particular importance in cases of BCNE, and where possible, molecular techniques should be employed on explanted tissues in these cases.

#### 5.2.1. Gram Staining and Culture

Valve tissue has historically been sent for Gram staining and culture: Positive culture from a cardiac vegetation is considered a major pathological diagnostic criterion in the modified Duke criteria [2]. Despite this, it should be noted that culture of valve tissue has low sensitivity and specificity, with positive cultures noted in only 6% to 26% of cases [11,45]. In cases where a cardiac device is extracted, the device itself should be submitted for sonication [46].

#### 5.2.2. Histopathology

Conventional hematoxylin and eosin (H&E) stains confirm the presence of inflammation and necrosis and can provide insight into the acuity of the infectious process. Specialized stains such as methenamine silver (fungi), acid-fast (mycobacterial), Warthin–Starry silver (*Bartonella* spp.), and Periodic acid–Schiff (*T. whipplei*) can be used to detect specific organisms in the appropriate clinical setting [47]. Histopathological analysis may also aid with the diagnosis of non-infectious causes of endocarditis, including auto-immune and neoplastic causes.

#### 5.2.3. Molecular Techniques

The molecular methods described in relation to blood and plasma specimens may also be performed on explanted valve tissue. As bacterial DNA is generally more abundant in valve tissue versus whole blood or plasma [48], where valve tissue is available, molecular techniques should be performed on these samples. In patients where serology is positive or equivocal, specific PCR assays may be used to confirm infection, e.g., *Bartonella* or *Brucella* sp. *Bartonella* PCR from valve tissue has been found to have a sensitivity of 92% vs. 33–36% from whole blood and plasma, respectively [29]; the sensitivity for other organisms has not been defined. Targeted sequencing of explanted valve tissue with 16S rRNA has been employed widely over the past decade. In cases of BCNE, an organism was identified in valve tissue in 60–100% of cases across five studies [49,50,51,52,53]. With respect to sMGS on cardiac valve tissue, Flurin et al. reported a sensitivity of 85.9–100% and a specificity of 72.7–100% based on two retrospective cohort studies and one prospective cohort study. In one of these studies, a pathogen was identified on explanted valve tissue in 21/21 patients with both blood and valve culture-negative endocarditis [54].

### 5.3. Suggested Diagnostic Approach

A suggested algorithm for the diagnosis of BCNE is outlined in Figure 1. Blood cultures should be obtained prior to administration of antibiotics, as outlined above in all patients. Where blood cultures remain negative at 48 h, the laboratory should be directed to prolong incubation of blood cultures for 14 days and serologies for *Bartonella* sp. and *C. burnetii* should be obtained. Serologies for *Brucella* sp., auto-immune diseases, and plasma PCR for *T. whipplei* can be considered on a case-by-case basis. Where surgery is not planned, broad-range bacterial PCR or shotgun metagenomic sequencing should be performed on whole blood where possible.

In patients who proceed to surgery, valve tissue should be sent for histopathological examination; specific stains can be performed as guided by the appearance of the resected valve and the clinical scenario. Fresh excised tissue should also be held for molecular testing, and molecular testing should be prioritized over Gram staining and culture where a microbiological diagnosis has not been established. Specific PCR assays can be performed in cases of equivocal serum serologies or where the history is suggestive of a zoonotic agent. At present, there is insufficient evidence to recommend tMGS over sMGS, and molecular testing should be guided by local availability of diagnostic techniques. In all cases where next-generation sequencing is performed, it is essential to correlate results with the clinical context.

## 6. Therapy

Treatment for BCNE depends on the likely causative agent and the presence of prosthetic material. Surgery is frequently required, and surgical indications are outlined in the 2015 American Heart Association guidelines [1]. In cases of acute native valve BCNE, an empiric therapy regimen providing coverage for *S. aureus*, beta-hemolytic streptococci, and aerobic Gram-negative bacilli is indicated. Dual therapy with vancomycin and cefepime is an appropriate choice and should be continued for 6 weeks [1]. In cases of subacute native valve BCNE, coverage of infection due to *S. aureus*, viridans group streptococci, HACEK organisms, and enterococci is indicated, and therapy with vancomycin and ampicillin–sulbactam is an appropriate regimen [1].

Therapy of blood culture-negative prosthetic valve endocarditis (PVE) should be determined based on the timing of the infection. Where onset of IE is within one year of surgery, antimicrobial therapy should provide coverage for staphylococci, enterococci, and aerobic Gram-negative bacilli. An empiric regimen with vancomycin, gentamicin, cefepime, and delayed addition of rifampin (after three to five days of therapy) is appropriate. Rifampin has historically been recommended specifically in cases of staphylococcal prosthetic valve endocarditis, although recent data have called this into question [55]. For cases of PVE occurring ≥1 year following surgery, antimicrobial therapy for coverage of infection due to staphylococci, viridans group streptococci, enterococci, and HACEK organisms is indicated. A typical empiric therapy is vancomycin and ceftriaxone [1].

If there is a reasonable chance of infection due to a zoonotic agent, empiric therapy versus the suspected agents can be started. Directed therapy for zoonotic agents and fastidious organisms is outlined in Table 1. Owing to the lack of large series, the optimal duration of treatment of IE due to these pathogens is unknown. The treatment of *C. burnetii* endocarditis bears mention given the inclusion of hydroxychloroquine along with doxycycline as part of the suggested treatment regimen. As *C. burnetii* replicates within macrophages and monocytes where the acidified phagosomal compartment decreases the bactericidal efficacy of antibiotics, it has been proposed and subsequently validated in retrospective case series that an alkalizing agent such as hydroxychloroquine may improve outcomes [56,57]. If hydroxychloroquine is not tolerated, co-treatment with a fluoroquinolone has been shown to be equally as efficacious [14].

## 7. Prognosis

Kong et al. conducted the largest study to date comparing outcomes of culture-positive and culture-negative endocarditis using a large European registry to obtain 1-year follow-up data on 3113 patients [58]. One-year mortality was significantly higher in the BCNE group (hazard ratio (HR) 1.28, 95% confidence interval (CI) 1.04–1.56). Notably, however, 1-year mortality was not significantly different in the cohort of patients who underwent surgery in the culture-negative and culture-positive groups, and in the BCNE group, surgery was significantly associated with survival (HR 0.49, 95% CI 0.41–0.58).

## 8. Conclusions

Blood culture-negative endocarditis remains a clinical challenge and is associated with worse outcomes than culture-positive endocarditis. The diagnostic landscape has evolved with the emergence of non-culture-based methodologies, including serological assays, targeted metagenomic sequencing, and shotgun metagenomic sequencing. These approaches have enabled the identification of pathogens that were historically challenging to detect through conventional blood cultures, and their role is likely to assume greater prominence as these technologies are validated and become more widely available outside of reference laboratories. The diagnostic algorithm outlined in this review provides a comprehensive and rational approach to the evaluation of BCNE cases, emphasizing the importance of thorough clinical history, appropriate blood culture techniques, serological testing, and the judicious use of molecular methods on both blood specimens and explanted cardiac tissue.

## Figures and Tables

**Figure 1 pathogens-12-01027-f001:**
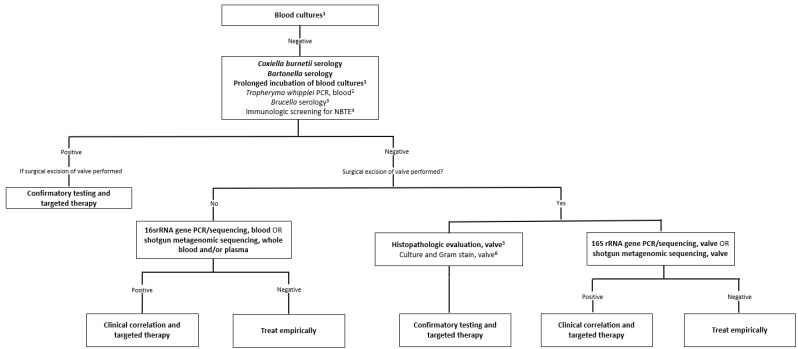
Diagnostic algorithm for the identification of the microbiological etiology of infective endocarditis. Adapted from Liesman et al. [33]. The algorithm is intended for use in patients with definite or possible infective endocarditis based on the modified Duke criteria. Strong recommendations appear in boldface. ^1^ Details related to the appropriate collection and incubation of blood cultures are included in the text. ^2^ The sensitivity of *T. whipplei* PCR from blood in endocarditis is unknown; a negative result should not be used to rule out *T. whipplei* endocarditis. ^3^ *Brucella* serology should be performed routinely in endemic regions or where the patient has specific risk factors (see Table 1). ^4^ Consider autoantibodies and work-up for malignancy as detailed in the text. ^5^ Histopathologic evaluation is used to evaluate for infectious and noninfectious etiologies and for correlation with microbiology test results. Subsequent directed testing may include specialized stains e.g., PAS-D staining for *T. whipplei*, or specific PCR assays e.g., *Bartonella* sp., *Coxiella burnetii*, *Cutibacterium acnes*. ^6^ If sufficient valvular tissue is available after sampling for histopathological and molecular (microorganism-specific and broad-range) testing, consider culture and microbiology Gram stain. Due to the low sensitivity and specificity of culture, molecular testing should be prioritized over culture.

**Table 1 pathogens-12-01027-t001:** Blood culture-negative endocarditis: diagnostic clues and treatment.

Organism	Risk Factor(s)/Diagnostic Clues	Diagnostic Tests	Suggested Treatment
*Coxiella burnetii* (Q Fever)	Inhalation of aerosols from infected animals (cattle, dogs, cats), Ingestion of unpasteurized dairy products, bioterrorism	*Coxiella burnetii* antiphase I IgG Ab titer > 1:800 *Coxiella* specific PCR on blood or valve tissue	Doxycycline + Hydroxychloroquine (18 months) [14]
*Bartonella henselae*	Contact with cats	Indirect immunofluorescence assays for detection of IgM and IgG antibodies to *Bartonella* spp. with IgG titer ≥ 1:800 *Bartonella* specific PCR on blood or valve tissue	Gentamicin IV (2 weeks) + Ceftriaxone IV (6 weeks) [1]
*Bartonella quintana*	Presence of body lice, contact with homeless shelters		
*Tropheryma whipplei*	Exposure to soil or farm animals	*Tropheryma whipplei* specific PCR on blood or valve tissue	Penicillin G or Ceftriaxone IV (2–4 weeks) followed by co-trimoxazole for one year [15]
*Brucella* sp.	Contact with unpasteurized dairy products, undercooked meat, or infected farm animals (sheep, cattle goats). Travel to endemic regions: the Mediterranean basin, the Middle East, Mexico [16]	Blood cultures *Brucella* total antibody titer ≥ 1:160 by standard tube agglutination test *Brucella*-specific PCR on blood or valve tissue	Gentamicin IV (4 weeks) followed by Rifampin and doxycycline (12 weeks) [17]
*Legionella* sp.	Exposure to artificial water systems	Molecular methods (targeted or shotgun metagenomic sequencing)	Macrolide + rifampin/ciprofloxacin (6 weeks)
*Mycoplasma hominis*	History of vaginosis or pelvic inflammatory disease	Molecular methods (targeted or shotgun metagenomic sequencing)	Doxycycline (4–6 weeks) [18]
Fungi	Intravenous drug use, organ transplantation, indwelling catheter, HIV positive	Blood cultures (*Candida* sp.), fungal blood cultures Shotgun metagenomic sequencing	Prolonged therapy based on species identified and susceptibility data
Tuberculosis *Mycobacterium chimera*	Tuberculosis exposure Cardiac surgery [19]	Mycobacterial cultures, histopathology with Ziehl–Neelsen stain Shotgun metagenomic sequencing	Prolonged therapy based on species identified and susceptibility data
Non-infective endocarditis			
Behcet’s Disease	Young male, aortic insufficiency, recurrent oral and genital ulcers	Clinical diagnosis, positive pathergy test	Immunosuppression ± anti-coagulation
Lupus endocarditis	Female patient, rash, cytopenias, arthralgias, kidney injury, history of autoimmune disease	Serologies and clinical correlation	Immunosuppression ± anti-coagulation
Marantic endocarditis	Known primary malignancy	Tumor markers, cancer screening	Cancer-directed treatment, anticoagulation
Allergic endocarditis on porcine bioprosthesis	Multiple small vegetations, allergy to porcine products	Clinical diagnosis	Replacement with non-porcine bioprosthesis

IgG Ab: immunoglobulin G antibody; PCR: polymerase chain reaction; IgM Ab: immunoglobulin M antibody.

## Data Availability

No new data were created or analyzed in this study. Data sharing is not applicable to this article.

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
