# Peer review of "Updates in Culture-Negative Endocarditis"

_pathogens, 2023, doi:10.3390/pathogens12081027_

Round 1

Reviewer 1 Report

In this paper, the authors conducted an updated and comprehensive review on culture-negative endocarditis. This paper contains a systematic tune-up on this topic of utility to the clinical cardiologist. The length of the text is appropriate as are the references.  

SPECIFIC COMMENTS Figure 1 is shown twice

Author Response

The authors thank the reviewer for the below comments. Figure 1 has been updated in the revised manuscript. A number of other revisions have been made as per suggestions from peer review. Please see the revised manuscript with revisions.

Reviewer 2 Report

The authors reviewed the definition, epidemiology, diagnostic approach, therapeutic strategy, and prognosis of blood culture-negative infective endocarditis.

The manuscript is well-written, and I found no critical issues except for a few typos in abbreviations as follows.

5.1.3. Molecular techniques 

microbial cell-free DNA (mcfDNA) 

5.2.3. Molecular techniques 

Targeted sequencing of explanted valve tissue with 16S rRNA 

Figure 1

16srRNA gene PCR -> 16S rRNA gene PCR

Author Response

The authors thank the reviewer for the below comments. Typos have been corrected. A number of other revisions have been made as per suggestions from peer review. Please see the revised manuscript with revisions.

Reviewer 3 Report

A study from a world-renowned centre on a topic that is still current and at the same time a major clinical problem. Despite the passing years, the development of new diagnostic methods, we still encounter problems with reliable identification of the pathogen. The very bold title of the analysis "Culture-Negative Endocarditis" makes us realize and remind us of the simple truth that a patient can suffer from infective endocarditis despite not meeting the modified Duke criteria and that the mortality rate in the FU period of patients with culture-negative endocarditis is not lower and even higher. And that failure to meet the modified Duke criteria does not rule out this potentially fatal disease.

The authors clearly presented modern diagnostic possibilities, which unfortunately are not yet available in smaller centers. The authors focused on the classic infective endocarditis, however, the development of methods of controlling the heart function using implantable electronic devices (CIED) contributed to creation of a previously non-existent variant of this disease - CIED-related endocarditis, in which diagnostic problems tend to be similar. The issue fully deserves a similar study, which I strongly encourage such competent authors to do. The issue of CIED-related culture-negative endocarditis cannot be analyzed in a few paragraphs (other literature), therefore I do not suggest extending the article.

With some review comments. Carefully check the Pathogens Regulations (MDPI) for the presentation of cited references. As far as I remember, the issue number and doi number should not be given. You should also check the recommended number of authors of cited publications.

Author Response

The authors thank the reviewer for the below comments. The reference section and style have been updated. A number of other revisions have been made as per suggestions from peer review. Please see the revised manuscript with revisions.

Reviewer 4 Report

Please see the file attached.

English language is fine. A few minor issues have been pointed out to the authors in the attached comments.

Author Response

The authors thank the reviewer for the below comments. The introduction and conclusion sections have been rewritten. The typos identified by the reviewer have been corrected (highlighted in yellow). A number of other revisions have been made as per suggestions from peer review. Please see the revised manuscript with revisions.